# Chiral high-harmonic generation and spectroscopy on solid surfaces using polarization-tailored strong fields

Tobias Heinrich [1], Marco Taucer[2], Ofer Kfir [1,3], P. B. Corkum[2], André Staudte [2], Claus Ropers[1,3] & Murat Sivis [1,3] ✉

Strong-field methods in solids enable new strategies for ultrafast nonlinear spectroscopy and provide all-optical insights into the electronic properties of condensed matter in reciprocal and real space. Additionally, solid-state media offers unprecedented possibilities to control high-harmonic generation using modified targets or tailored excitation fields. Here we merge these important points and demonstrate circularly-polarized high-harmonic generation with polarization-matched excitation fields for spectroscopy of chiral electronic properties at surfaces. The sensitivity of our approach is demonstrated for structural helicity and termination-mediated ferromagnetic order at the surface of silicon-dioxide and magnesium oxide, respectively. Circularly polarized radiation emanating from a solid sample now allows to add basic symmetry properties as chirality to the arsenal of strong-field spectroscopy in solids. Together with its inherent temporal (femtosecond) resolution and non-resonant broadband spectrum, the polarization control of high harmonics from condensed matter can illuminate ultrafast and strong field dynamics of surfaces, buried layers or thin films.

---

[1] 4th Physical Institute–Solids and Nanostructures, University of Göttingen, Göttingen, Germany. [2] Joint Attosecond Science Laboratory, National Research Council of Canada and University of Ottawa, OttawaON, Canada. [3] Max Planck Institute for Biophysical Chemistry, Göttingen, Germany. ✉email: murat.sivis@uni-goettingen.de

Nonlinear spectroscopy has made a huge leap forward with the proof of high-harmonic generation (HHG) in condensed matter[1–4]. Recently, numerous excellent studies, involving band reconstructions[5], exciton analysis[6], momentum dependent phases[7–9], and valence-electron mapping[10] have shed new light on various solid-state phenomena. At the same time, the functionalization of solid targets through structural and chemical modifications[11–13] or nano-confined and tailored excitation fields[13–16] has led to unprecedented possibilities for the controlled generation of high harmonics. The variation of the excitation field's polarization, in particular, can provide access to crystal orientation-dependent information[17–20]. When the crystal symmetry is known, the study of phenomena in the sample can be conducted by combining linearly or circularly polarized excitation with a polarization analysis of the harmonic emission. Moreover, in such schemes, the generation of harmonic radiation with elliptical polarization is possible[20–22]. The use of nontrivial tailored driving fields can significantly enrich solid-state HHG by providing a universal control of the high-harmonic polarizations in arbitrary (symmetric) crystal structures, which is key to efficient generation of circularly polarized radiation and to symmetry-resolved chiral spectroscopy.

In HHG from atoms and molecules, intricate field symmetries already proved their important role, including the probing of orbital angular momentum states with symmetries across the laser wavefront[23–25] and symmetries at the level of local electric fields[26–28]. Of great interest for crystalline solids are bi-chromatic light fields with controllable polarization, which can possess discrete rotational symmetries, and are known from circularly-polarized HHG in gases[29,30]. In the particular case of a bi-circular field comprising counter-rotating fundamental wavelength and its second harmonic, the rotational symmetry is threefold, and the field's angular momentum is imprinted on the generated harmonics. In solids, where the HHG is affected by the crystalline symmetry, such tailored excitation fields would allow for a symmetry-sensitive probing of chiral surface-band features, which is particularly relevant for the study of correlated electronic systems or magnetic properties.

Here, we demonstrate that bi-chromatic rotational symmetric driving fields[29–31] probe rotational and chiral symmetries at the surfaces of bulk-insulating crystals via HHG. In particular, we match a threefold driving field with threefold, fourfold, and sixfold structures of specific crystal cuts of silicon dioxide (quartz) and magnesium oxide (MgO) and find a high sensitivity of circularly-polarized HHG in solids to structural helicity and surface magnetism. By evaluating the difference in the resonant, i.e., near-band-edge, absorption between left- and right-circular polarized harmonics we can measure chirality-mediated band shifts in the respective crystal systems. While chiral band shifts in quartz(0001) and at the polar MgO(111) oxygen-terminated surface[32] is expected due to the screw-like structural helicity (P321 space group)[33] and the reconstruction-mediated ferromagnetism at the polar surface[34,35], respectively, we also discover a chiral footprint also on the cubic non-polar surface of MgO(100).

## Results

**HHG in solids with bi-circular driving fields**. In this study single-crystalline quartz and MgO targets are excited under moderate vacuum conditions ($< 10^{-6}$ mbar pressure) with bi-circular two-color laser pulses (50 fs pulses at 1 kHz repetition rate) comprising a circularly-polarized fundamental field at 800 nm wavelength and a counter-rotating circularly-polarized second harmonic (400 nm wavelength). The field strengths we used (10.5 V nm$^{-1}$ in quartz and 7.7 V nm$^{-1}$ in MgO) are comparable to other studies[4,17] that observed strong-field HHG in these materials. Figure 1a schematically illustrates the experimental setup, where high-harmonic radiation is generated in reflection geometry from a crystal surface and collected with an extreme-ultraviolet spectrometer. The driving bi-circular field exhibits a threefold helical polarization (see Supplementary Fig. S1 for details), which we utilized as a spectroscopic probe via circular HHG. Figure 1b depicts this principle, which involves rotational symmetry probing by changing the angle $\theta$ between the field and the crystal as well as chiral sensitivity through the circular polarization state of the generated harmonics. A typical spectrum spanning from the fifth harmonic (7.75 eV photon energy) to the tenth harmonic (15.5 eV) is shown in Fig. 1c for quartz(0001). The spectra for all crystals are shown in Supplementary Fig. S2. The near-perfect suppression of every third harmonic is the selection rule that corresponds to circular polarization of the unsuppressed harmonics[29,30] (see Supplementary section selection rules). More precisely, the circular

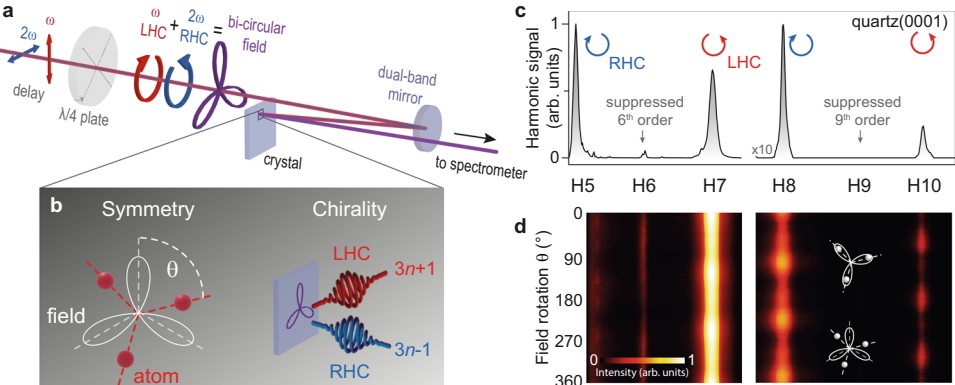

**Fig. 1 Chiral high-harmonic generation in solids. a** Two-color laser pulses (frequencies marked $\omega$ and $2\omega$) with perpendicular polarization and adjustable phase delay can be converted to circularly-polarized fields with opposite helicity using a dichroic quarter-wave plate with the fast-optical axis set to 45° relative to the laser polarization. The superposition yields a bi-circular field with a threefold rotation symmetry. The ellipticity and the orientation of the field (see angle $\theta$ in (**b**)) depend on the quarter-wave plate rotation and the phase delay between the two spectral field components, respectively. The emitted harmonic radiation is collected with an extreme-ultraviolet spectrometer. LHC left-handed circular, RHC right-handed circular. **b** Schematic depiction of the symmetry and chirality probing principle using a bi-circular field for HHG in solids. **c** High-harmonic spectra from quartz(0001) (harmonic orders label with H5–H10), showing suppression of every third harmonic order. The blue and red arrows indicate the helicity of the harmonics. **d** Spectrogram for quartz (0001) as a function of the threefold field rotation angle $\theta$ (see also pictograms), exhibiting a threefold beating.

polarization of the harmonic orders $q = 3n + 1$ (4, 7, 10, … ) exhibits the same sense of rotation as the fundamental driving field while the orders $q = 3n - 1$ (5, 8, 11, … ) have an opposite polarization helicity (see Fig. 1a–c). For a harmonic generation in isotropic targets, such as gases, the radiation symmetries are dominated by the threefold polarization shape of the driving bi-circular field. Since threefold and sixfold rotation symmetries inherently conform to the threefold symmetry of the laser field, the same selection rules apply for the termination planes of quartz (0001) and MgO(111) (cf. spectrum in Supplementary Fig. S2). Conversely, crystals exhibiting other symmetries, e.g., the fourfold MgO(100), remove the constraints on the selection rules and lead to a less pronounced suppression of every third harmonic (see the spectrum in Supplementary Fig. S2 and supplementary information for details on the selection rules). On the other hand, the suppression of every third harmonic confirms the threefold symmetry of the driving field, which allows for the probing of the crystal axes by polarization rotations (see Fig. 1b left). The spectrogram in Fig. 1d shows allowed harmonic orders which peak every 120°, corresponding to the P3 point-group of the quartz crystal. We attribute the higher angular sensitivity, i.e., the stronger modulation of the signal, for increasing harmonic orders to the nonlinear intensity scaling of the generation process. More specifically, the angular resolution is determined by the spatial generation anisotropy of the harmonics (whether in the perturbative or non-perturbative regime) and the particular polarization contrast of the threefold excitation field (see Fig. S1a). While the influence of the underlying mechanism warrants further investigation, the observed phase differences between individual harmonics (cf. H8 and H10 in Fig. 1d) already suggest a non-perturbative generation of higher harmonics[3,5,36] in quartz, which could be relevant for the study of diverse solid-state phenomena[5,10,37].

**Inversion-symmetry probing.** The lack of inversion symmetry of bi-circular fields is particularly relevant for the probing of material inversions. Figure 2 analyzes the rotational dependence of harmonic generation driven by bi-circular fields and inversion-symmetric linear-polarized fields for different crystals. The solid targets quartz(0001), MgO(111) and MgO(100) represent threefold, sixfold, and fourfold rotational symmetries, respectively. The shown polar plots of the eighth harmonic intensity (ninth harmonic in MgO for linear polarizations) are extracted from the spectrograms (see also Supplementary Figs. S2 and S3). In quartz, an inversion-symmetry-broken crystal, the bi-circular laser field probes the threefold rotational symmetry and the crystal orientation accurately (Fig. 2a). Contrary, a polarization scan with a linearly-polarized field, which possesses a twofold symmetry, exhibits an ambiguous sixfold beating (Fig. 2d) corresponding to an inversion-symmetrized threefold rotation. For crystals with sixfold symmetry as MgO(111) (Figs. 2b, e) rotating the bi-circular or linearly polarized field yields similar sixfold patterns. In the case of the fourfold MgO(100) (Fig. 2c, f), the harmonic emission from the linearly-polarized fundamental exhibits the expected, fourfold pattern, while for excitation with the bi-circular field the signal loses the appearance of its fourfold symmetry. The convolution of the threefold field and the fourfold crystal should result in a 12-fold signal[38], which is not resolvable. Since in MgO(111) the angular-dependent emission peaks are narrow enough to resolve a 12-fold symmetry, we conclude that the MgO(100) is more isotropic. It may be possible to refine the angular modulation of the harmonic intensity by tuning the parameters of the driving field, such as the amplitude ratio of fundamental and second harmonic, which has an influence on the symmetry modulation of the field, as discussed in "Methods".

**Structural and magnetic circular dichroism in solid-state HHG.** Chiral electronic properties become apparent when comparing circular HHG for the left and right helicities of the bi-circular field. We find chiral spectral signatures in quartz and MgO and attribute those to structural helicity and ferromagnetic order, respectively. The difference of spectrograms for left and right bi-circular driving fields (see Supplementary Fig. S2) in quartz, shown in Fig. 3a, reveals a strong angle-dependent spectral shift of the seventh harmonic, as outlined in Fig. 3c (gray dots). A similar shift of the fifth harmonic is observed for both crystal cuts of MgO, which is exemplified in Fig. 3b, d for MgO(100). Importantly, regardless of the crystal termination planes in MgO (both 100 and 111, cf. Supplementary Fig. S4), the spectral shifts are independent of the field rotation angle. In both materials, the harmonic orders exhibiting a spectral shift are located near the bandgap energy and we refer to these harmonic orders as "on-resonance" in the following.

Our observations can be explained by circular dichroism, i.e., a helicity-dependent absorption of generated harmonic radiation. This absorption leads to red- and blue-shifted on-resonance harmonics having left- and right-circular polarization, respectively (see red and blue dots in Fig. 3e, f). This effect is similar to dichroic absorption measurements with external sources, however here, the probe emanates from the sample itself and is sensitive to the crystal symmetry. To evaluate the chiral spectral response, we consider near-band-edge absorption of the opposite circularly-polarized harmonics, with different helicity-dependent absorption channels in both materials. In quartz, the screw-like crystal arrangement possesses a structural chirality (see schematics for the $3_1$ and $3_2$ structure in Fig. 4a), where the two enantiomers exhibit different band structures[33], as qualitatively drawn in Fig. 4b. Hence, opposing helicities of emitted harmonics probe a different band structure in $3_1$ and $3_2$ (see purple arrows in the band diagram in Fig. 4b). On the other hand, in MgO—being an achiral crystal—the surface ferromagnetism dominates the helicity-selective absorption. Spontaneous symmetry-breaking in the non-stoichiometric surface reconstructions[32,35,39] lead to the formation of a ferromagnetic layer with a spin-polarized metallic surface[39,40]. The oxygen termination of the polar MgO(111) may

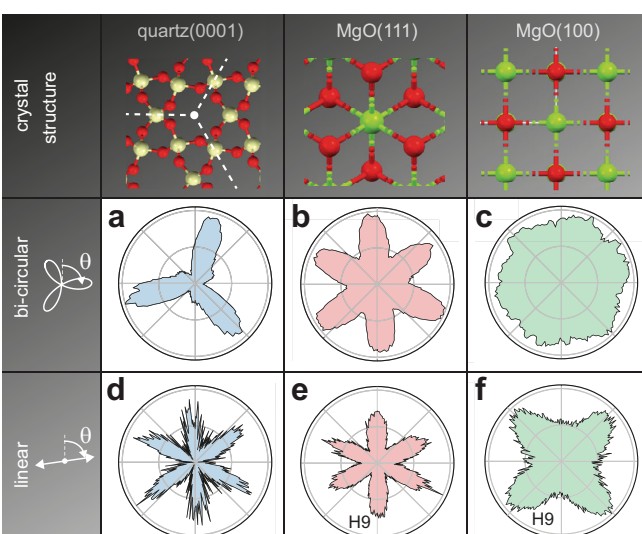

**Fig. 2 Bi-circular and linear polarization rotation scans for quartz and MgO. a–c** The yield of the eighth harmonic order (H8) as a function of the rotation angle of the threefold field. **d–f** Yield of H8 as a function of linear polarization angle. For MgO, H8 from linearly polarized drivers is symmetry-forbidden, hence H9 is presented.

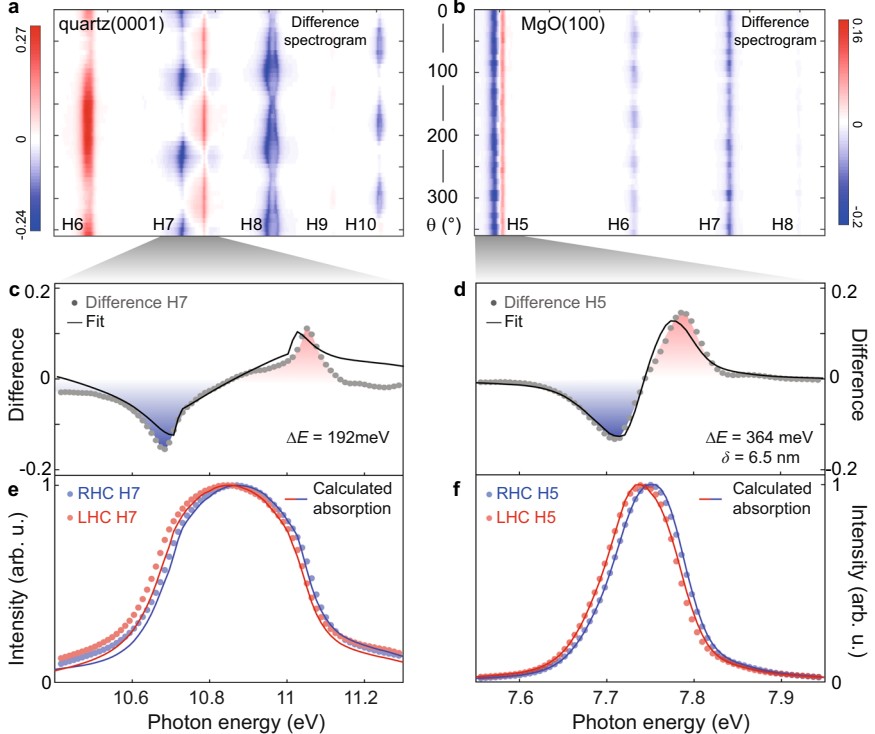

**Fig. 3 Sensitivity of HHG to structural helicity of quartz and surface-ferromagnetic order of MgO. a, b** Difference signal of two intensity-normalized spectrograms recorded with opposing helicities of the driving field generated in (**a**) quartz(0001) and (**b**) MgO(100). For single helicity spectrograms cf. Fig. 1d and Supplementary Fig. S2. **c, d** Lineouts of the on-resonance harmonic difference signal in (**c**) quartz and (**d**) MgO. The solid lines represent a fit based on calculations of the circular dichroism (fit parameters are the chiral energy splitting $\Delta E$, and the effective interaction length $\delta$). **e, f** Normalized intensity of right-handed-circularly (blue, RHC) and left-handed-circularly (red, LHC) polarized harmonics in (**e**) quartz and (**f**) MgO. The solids lines represent the calculated circular dichroism. Lineouts are taken at an angle of $\theta = 120°$ in the case of quartz and averaged over all angles for MgO.

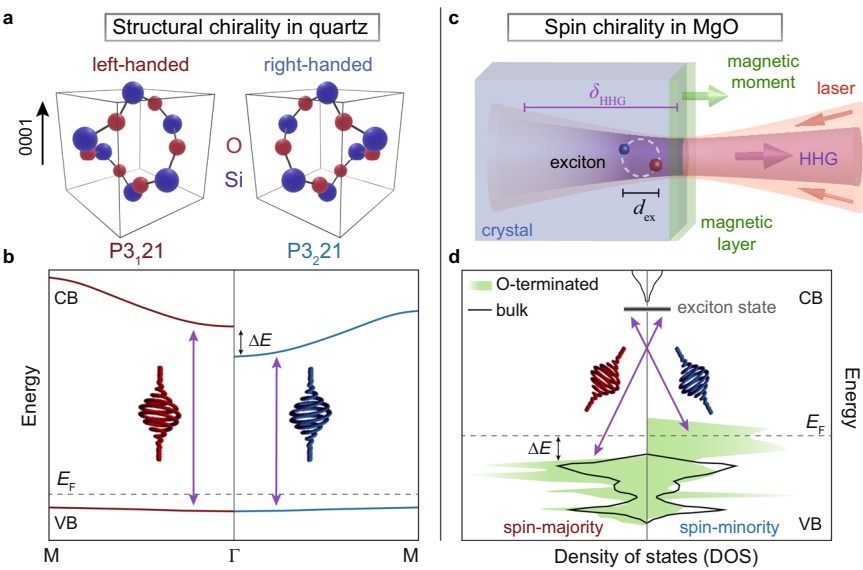

**Fig. 4 Chirality in quartz and MgO. a** Schematic of left- and right-handed enantiomers ($P3_121$ and $P3_221$) of quartz(0001). **b** Qualitative band structure of the two quartz(0001) enantiomers, adapted from ref. [33]. **c** Schematic of the high-harmonic generation (HHG) in MgO. Near-surface excitons with exciton diameter $d_{ex}$ may couple to the surface magnetic moment. Harmonic emission generated within the absorption length $\delta_{HHG}$ is detected in our experiment. **d** Simplified surface density of states (DOS) for the oxygen-terminated MgO(111) surface (green, see also inset for crystal structure) and bulk MgO (black), adapted from ref. [39] (VB valence band, CB conduction band, $E_F$ fermi energy, $\Delta E$ band splitting). The purple arrows mark the excitations with opposite helicities of circularly polarized light.

create a magnetic moment perpendicular at the surface[41], as schematically depicted in Fig. 4c. The band structure for the minority and majority spin-polarized charge carriers on the surface of MgO(111) is drawn in Fig. 4d. The annotated arrows mark the different absorption for left- and right-circular polarized harmonics (see purple arrows in Fig. 4d). In MgO(100), magnetic ordering may arise from vacancies and impurities or faceted surface reconstructions[40,41], while bulk ferromagnetism can be excluded at room temperature[35]. The fifth harmonic is sensitive to the ferromagnetism at the MgO surface due to its absorption by excitons, allowing for a penetration depth of only $\delta_{\mathrm{HHG}} = 15$ nm (ref. [42]). This attenuation length is comparable to the electron-hole binding distance $d_{\mathrm{ex}}$ (about 6 nm in MgO (ref. [43])), which highlights the significance of these surface-localized excitons in our measured spectra.

In order to reproduce the spectral response in Fig. 3, we calculate the helicity-dependent absorption of harmonics with left- and right-circular polarization by considering energetically-shifted absorption coefficients $k(E \pm \Delta E/2)$ for quartz[44] and MgO (ref. [42]) The term $\pm \Delta E/2$ represents the energy shift for opposite circular polarizations (see Supplementary Fig. S5). While in the calculation for quartz we consider the entire optical penetration depth (see Eq. (1) in "Methods"), in the case of MgO, we introduce an effective interaction length $\delta$ as an additional parameter (see Eq. (2) in "Methods") to model a surface sensitivity. We fit the absorption spectra shown in Fig. 3e, f to Eq. (1) and (2) by varying $\Delta E$ (and $\delta$, in the case of MgO), and present the final curve alongside the difference spectra in Fig. 3c, d (solid lines). In both materials, the absorption model yields significant spectral shifts for the on-resonance harmonic orders while the other harmonic orders show only intensity differences, similar to what we observe in Fig. 3a, b. This is consistent with the derivative of the absorption coefficient being largest at the band edge, which justifies the extraction of the band splittings from the on-resonance harmonic signals. The obtained band splitting values of $\Delta E = 192$ meV in quartz (maximum, at 215° field rotation), $\Delta E = 364$ meV in MgO(100), and $\Delta E = 395$ meV in MgO(111) (average over all field rotation angles), are in good agreement with the theoretically predicted changes of the band energy[33,39,45] (cf. Fig. 4b, d). Furthermore, the obtained interaction length of $\delta = 4 - 7$ nm in MgO matches the expected exciton diameter, which suggests a coupling of the surface-magnetic moment to exciton states. The intrinsically different origin of the circular dichroism in quartz and MgO is revealed by the remarkably different dependence of their band splitting on the field-rotation angle, as shown in Fig. 5. In quartz (Fig. 5b), the band splitting is rotationally anisotropic with a threefold periodicity, indicating a crystal-axis-dependent band structure[33].

On the other hand, the isotropic band splitting in MgO (Fig. 5a) can be attributed to the ferromagnetic order on the surface[35]. The accuracy in which the chiral excitations on the surface explain our observations suggests that further theoretical and experimental analysis using tailored strong-fields under consideration of the energy-dependent emission phases (see Fig. 1d) could provide additional microscopic information on chiral systems, possibly even on the atomic level[10].

## Discussion

In conclusion, we demonstrate a distinct symmetry-dependent chiral sensitivity of high harmonics generated from bi-circular laser fields near crystal surfaces. We show that the tailored symmetry and angular momentum of the driving field polarization can be utilized to generate circularly-polarized harmonic radiation in solids with arbitrary crystal structures and allow for symmetry-resolved chiral spectroscopy of the generation medium. Specifically, the harmonic yield depends on the matching between the Lissajous polarization curve and the crystalline axes, which can be harnessed for the detection of inversion symmetry. The circular polarization of the generated harmonics enables probing of the crystalline chirality in quartz and ferromagnetism on the surface of MgO. Using harmonics at the band resonance, we extract the symmetry-resolved energy shearing of helicity-dependent (quartz) and spin-polarized (MgO) bands with a fitting model incorporating a polarization-dependent absorption coefficient. Compared to dichroic absorption spectroscopy with external light sources, the symmetry-sensitive, local generation as well as the large bandwidth and wavelength tunability of the harmonics represent clear advantages. Furthermore, the relaxed requirements on the vacuum conditions and the sample preparation allow for a wider range of materials to be studied. This is particularly relevant for insulating materials for which chiral spectroscopy on thin-film samples is possible but challenging with electron-based methods like spin-resolved photoemission spectroscopy[46,47] or spin-polarized scanning tunneling microscopy[48–50] due to surface charge and surface contamination over time[51]. Besides illustrating an all-optical scheme for surface-sensitive spectroscopy, our method also facilitates compact, gas-free extreme-ultraviolet light sources for experiments involving ultra-high vacuum. Moreover, extending the spectroscopic capabilities of our approach, the inherent femtosecond pulse duration of the harmonics and the breaking of expected selection rules imply a powerful, single-shot probe for ultrafast dynamics. In addition to the detection of global properties such as orientation, chirality, or phase transitions, setting the HHG on a particular resonance, e.g., in semiconductors[1,3,9,52] or insulators[4,10], will allow targeting other surfaces phenomena with the potential of direct or diffractive imaging[11]. Thus, this work opens a path for detailed ultrafast surface spectroscopy and microscopy of dielectric and insulating materials.

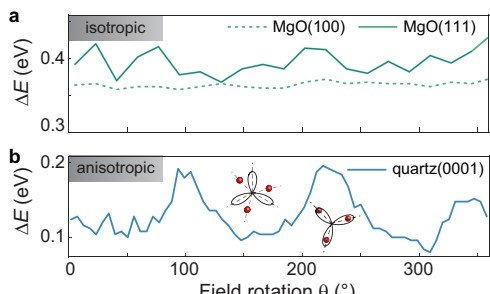

**Fig. 5 Chirality-mediated band shifts. a, b** Band shifts as a function of field rotation for MgO (**a**) and quartz (**b**). The orientation-independent energy splitting in MgO suggests that the chirality originates from spin-polarized surface states.

## Methods

**Crystals**. In our experiment we use commercially available (MTI crystals), single-crystalline silicon dioxide (quartz), and magnesium oxide (MgO) targets with 10 mm × 10 mm lateral size and 300 µm thickness. The quartz crystal is oriented along the 0001-direction perpendicular to the z-plane (space group P3$_1$21 or P3$_2$21). Two additional MgO crystals are cut perpendicular to the [111]- and [100]-direction, respectively.

**The laser system and two-color bi-circularly-polarized driving fields**. The laser system used in this study (Coherent Elite amplifier with Vitesse seed oscillator) delivers milli-Joule-level, 50 femtoseconds (FWHM) pulses with 800 nm central wavelength (see the red spectrum in Supplementary Fig. S1b) at a 1 kHz repetition rate. We are able to control the excitation power with a variable attenuator and the incident polarization with a half-wave plate.

Bi-circular–polarized two-color light fields are generated in an in-line MAZEL-TOV (MAch-ZEhnder-Less for Threefold Optical Virginia spiderwort)

apparatus. Supplementary Fig. S1a depicts the setup, which consists of a 40 cm plano-convex lens and a BBO crystal for the generation of second-harmonic laser pulses with 400 nm wavelength (see the blue spectrum in Supplementary Fig. S1b) and a polarization perpendicular to the fundamental polarization direction. Two thick horizontally counter-rotational tiltable calcite plates control the timing (overlap) of the two pulses and a dichroic quarter-wave plate converts the linear polarized pulses to oppositely circularly polarized fields. This configuration allows for the generation of bi-circularly-polarized two-color laser pulses with a defined relative phase, which leads to a threefold symmetric field distribution with fixed orientation in space. Therefore, the quarter-wave plate's fast optical axis is set to ±45° relative to the linearly polarized fundamental where the sign determines the direction of rotation. A wave plate angle of 0° leaves the perpendicular linear-polarizations of the two pulses unchanged. An additional tiltable thin calcite plate mounted on a rotational stage is placed before the quarter-wave plate, in order to fine-tune and scan the relative phase of the fundamental and the second-harmonic pulses. The phase delay between the fundamental and the second-harmonic pulses controls the relative rotational orientation of the threefold field distribution to the crystal axis (compare angle $\theta$ in Fig. 1). More details on the MAZEL-TOV apparatus and the generation principle can be found elsewhere[53]. Removing the MAZEL-TOV apparatus and the quarter-wave plate allows for the use of linearly-polarized laser excitation at the fundamental wavelength, with variable polarization angle via the half-wave plate.

In the experiment, the relative power of the second harmonic field to its fundamental was 6% for the measurements on quartz and 1% in the case of MgO. Considering the wavelength-dependent focusing of the 800 nm and 400 nm fields, the field amplitude ratios are 1:2 and 1:5 in the quartz and MgO measurements, respectively, which determines the actual shape of the driving fields in both cases (see Fig. S1a). Supplementary Fig. S1c, d show the threefold driving Lissajous curves for a field amplitude ratio of 1:2 for right- (c) and left- (d) helical rotation. An altered amplitude ratio has no impact on the rotational symmetry of the field nor the helicity of emitted harmonic radiation. It does, however, influence the spatial isotropy, i.e., the degree of angular modulation, of the field, which has an impact on the angle resolution in polarization scans.

We calculated the driving field under consideration of non-optimal quarter-wave plate retardance (see Fig. 1b). Therefore, we quantify the specified frequency-dependent retardance $R(\omega)$ with two linear fits at 400 nm and 800 nm wavelength. In addition, the spectrum of the driving pulse was measured and fitted by two Gaussians. Under the assumption of a Fourier limited pulse, we determined the frequency-dependent electric field $F(\omega)$. The time-dependent electric field can be calculated by a Fourier-transformation $F(t) = \mathfrak{F}[F(\omega)e^{-iR(\omega)}]$ and is compared to the field generated with a perfect quarter-wave plate. One optical cycle of the electric field $F(t)$ and its difference $\Delta F$ to an undistorted two-color field is shown in Supplementary Fig. S1c, d. The total driving field as well as the difference exhibit a threefold cloverleaf shape. We find that a left-helical field is accompanied by a right-helical residual component and vice versa (see Fig. S1c, d). As the residual field counter rotates a weak sixfold beating is expected when scanning the cloverleaf field orientation.

**Vacuum system and extreme-ultraviolet spectrometer.** We utilize a high-vacuum system at a base pressure of 1e−6 mbar for the HHG and spectral detection of the emitted extreme-ultraviolet radiation. A lens with a 40 cm focal length steers the laser through an optical window into the vacuum chamber. The beam is then retro-reflected of a dual-band dielectric mirror (Eksma, 052-4080-i0) under nearly normal incidence (horizontal and vertical incidence angles <1°) and focused to a focal spot size of 120 μm in diameter (full-width at half-maximum) on the target crystal surface. Due to the reflection geometry of the setup, the incident horizontal angle of the laser beam on the target is 5°, whereas the vertical incident angle is <1°. The generated high-harmonic radiation is collected with a focusing flat-field spectrometer (McPherson, model 234, grating with 1200 grooves/mm) and recorded with a phosphor-screen microchannel plate detector using a CCD camera.

**High-harmonic spectra and spectrograms.** High-harmonic spectra from the three crystal targets are shown in Supplementary Fig. S2a–c. The intensity of high-harmonics generated with the threefold bi-circular driving field (red) is compared to the signal generated with a two-color perpendicular-linearly polarized field (blue) to assess the suppression of individual harmonic orders. By measuring the spectra with bi-circular excitation as a function of the threefold field rotation θ we obtain the spectrograms, which are presented in Supplementary Fig. S2d–f. The spectrally-integrated signal of the eighth harmonic is used to generate the polar plots in Fig. 2a–c.

**Helicity-dependent splitting of on-resonance harmonic peaks.** The difference spectrograms in Fig. 3a, b and Supplementary Fig. S4 are determined from spectrograms, similar to those shown in Supplementary Fig. S2d–f, which are recorded for both helicities of the fundamental. The fifth harmonic generated in both facets of MgO and the seventh harmonic generated in quartz shows a spectral shift that stems from a chiral response in both materials systems.

We model the effect of magnetic circular dichroism in MgO and structural circular dichroism in quartz via fitting of a helicity-dependent absorption to the observed spectral shearing of the on-resonance harmonics. Considering energy splitting $\Delta E$ in the absorption channels for the two polarization states of the harmonics (cf. Fig. 4b, d) the helicity-dependent attenuation is described by energetically-shifted literature values of the absorption coefficient $k(E \pm \Delta E/2)$ (Supplementary Fig. S5). Here, the signs determine the circular polarization of the high-harmonic radiation. The respective fifth and seventh harmonics are close to the resonances in MgO and quartz, such that even small energetic shifts of $k(E)$ lead to a pronounced change in the absorption of an initial harmonic intensity $I_0(E)$. The intensity of shifted harmonics with left- and right-circular polarization ($I_{L,R}(E)$) then result from the following two equations for the respective crystal systems:

$$I_{L,R}^{\text{Quartz}}(E) = \int_0^\infty I_0(E)e^{-\frac{4\pi}{\lambda}xk\left(E \pm \frac{\Delta E}{2}\right)}dx = \frac{\lambda}{4\pi}I_0(E)k^{-1}\left(E \pm \frac{\Delta E}{2}\right) \quad (1)$$

$$I_{L,R}^{\text{MgO}}(E) = I_0(E)e^{-\frac{4\pi}{\lambda}\delta k\left(E \pm \frac{\Delta E}{2}\right)} \quad (2)$$

where $I_0$ is the initially unshifted harmonic peak and $\lambda$ is the wavelength of the harmonic. We used Eq. (1) and (2) to calculation the emitted harmonic intensity by estimating $I_0(E)$ as the mean of the recorded left- and right-circular polarized spectra. In the case of quartz, the spectrum was additionally fitted by a gaussian function to compensate for the saturated signal in the corresponding measurement. A helicity-dependent bulk effect is expected in quartz harmonics generated from every depth $x$ and weighted by their chiral absorption contribute to the emitted intensity. Contrary, in MgO we use an additional fit parameter $\delta$ to describe the interaction length in a near-surface region where the absorption is helicity-dependent. We assume that all generated harmonic radiation passes this surface near the region, neglecting the generation in the layer itself. This is justified since the major portion of the detected radiation is generated beyond this surface region as can be approximated by the penetration depth $\delta_{\text{HHG}}$.

In addition to a chiral material response, we also considered the energy splitting of excitonic states due to an optical stark effect from the residual field as a possible cause for shifted high-harmonics. Since changing of the fundamental helicity lead to inversion of all helicities including the residual fields (see Fig. S1c, d), we can conclude that no symmetry-breaking field is present and no stark shift signal is expected in our case[54,55]. Linear field components that result from the near-normal incidence of the bi-circular field on the crystal could lead to a symmetry-broken excitation. However, under consideration of the Fresnel coefficients, the calculated field values are many orders of magnitude too low for a significant optical stark effect.

## Data availability

The data that support the finding of this study are available from the corresponding author upon reasonable request.

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

## Acknowledgements

The authors thank Hugo Lourenço-Martins for helpful comments on the manuscript. This work was funded with resources from the Gottfried Wilhelm Leibniz Prize. O.K. gratefully acknowledges funding from the European Union's Horizon 2020 research and innovation program under the Marie Skłodowska-Curie grant agreement No. 752533. P. B.C. acknowledges funds from the United States Air Force Office of Scientific Research (AFOSR) under award numbers FA9550-16-1-0109.

## Author contributions

M.S. conceived and designed the experiment with contributions from M.T. and O.K. M. S., T.H., and M.T. conducted the experiments with contributions from O.K., analyzed the data, and prepared the manuscript with contributions from O.K. P.B.C., A.S., and C.R. All authors discussed the results and interpretation.

## Funding

## Competing interests

The authors declare no competing interests.
