## [Peer Review File · Nature Communications]

Reviewers' Comments:

Reviewer #1:

Remarks to the Author:

In this manuscript, the authors demonstrate a structural/chiral sensitivity of HHG at solid surfaces. The detection of inversion symmetry and the probing of the crystalline chirality are demonstrated with HHG driven by bi-circular laser fields. The results are novel and reliable. Since the manuscript successfully highlights a potential of HHG as an all-optical method for surface-sensitive spectroscopy, the contents would be interesting to the material science community as well as to the photonics community. There are, however, several concerns which should be clarified to consolidate the main claims of the authors. Please refer to the following comments.

1) Is the demonstrated sensitivity to the inversion symmetry specific to the strong-field regime or does it appear regardless of the excitation regime? The polarization rotation scans (as shown in Fig.2) for varied excitation intensity would provide the answer. Similarly, is the sensitivity to the crystalline chirality specific to the strong-field regime?

2) In Figure 1d, the authors display the polarization rotation scans for H5, H7, H8, and H10. Each order exhibits a three-fold beating, but with different offset phases. Please explain the reason why the offset phase varies with harmonic orders.

3) In the inversion-symmetry probing, does H8 provide any further information than what H5 provides? It is preferable that the authors could address the impact of using higher orders in the structural probing.

4) In addition to the spectral shift shown in Fig.3a and Fig.3b, there are also difference signals for H6, H8, and H10 in Fig. 3a and for H5 and H7 in Fig.3b. What is the origin of these difference signals?

5) Please provide the angle theta for the data shown in Fig.3c. The difference signal for H7 from quartz (0001) should be angle dependent according to Fig.3a.

6) One can imagine that linear circular dichroism measurement in the ultraviolet range would also detect the chiral properties. Please provide any advantage of the chirality detection based on HHG.

Reviewer #2:

Remarks to the Author:

The authors use a two-color excitation with two laser pulses having an opposite circular polarization to measure high-harmonic (HH) emission of solids as function of direction/chirality of the used bi-chromatic field with respect to crystal. The HH signal is demonstrated to track crystal symmetries of three different solids.

The work is an interesting extension of an earlier work, Nat. Photon 11, 227 (2017) by Huber group where linearly polarized light was already shown create distinctive direction and polarization sensitive HH as function of crystal orientation. The connection to this earlier work is neither referenced nor compared to determine the novelty or uniqueness of the present work. Especially, it is not clear which are the truly new aspects proposed in this submission. For example, the conclusion "we demonstrate a unique structural and chiral sensitivity of high harmonics generated from bi-circular laser fields near crystal surfaces" seems an over statement because many aspects of bi-chromatic spectroscopy can also be accessed by other forms of HH spectroscopy.

The experiment shown here are obviously high level, but before the unique aspects of bi-chromatic spectroscopy are articulated and demonstrated, the work remains more a technical extension of

earlier works. Publication in Nature Communication would require that the authors articulate and demonstrate a unique, important aspect that their approach adds to the HH spectroscopy. In its current form, this manuscript presents more of a technical, incremental extension to prior works.

To help the revision process, here are few technical comments:

Lines 39 & 56: It would be beneficial to elaborate the key idea and consequences of "bi-chromatic light fields" with few sentences for a non-expert.

Lines 80-82: Statement "The near-perfect suppression of harmonic orders $q = 3n$ (3,6,9,...) testifies to the circular polarization of the unsuppressed harmonics" is unclear for a non-expert.

Fig 1a: What is the role of field strengths (besides the phase delay) of two pulses to create the desired bi-circular field?

Figs 1d & 2: How unique are the crystal-direction dependencies to bi-circular approach? It seems many of the key features reported can be extracted by analyzing the details of a linear excitation. This approach is much simpler than bi-circular, and it has been demonstrated before. Or is the message that one should use few excitation schemes to resolve crystal symmetries? If yes, can you propose a combination of excitation schemes that works to characterize most crystals?

Line 119: Statement "A linearly-polarized field has an equal amplitude under inversion, with an opposite sign" is correct but probably will not open to a non-expert.

Line 127: Why is the 12-fold symmetry not resolvable by the used setup?

Line 145: Caption text "Difference signal of two intensity-normalized spectrograms" does not specify clearly which two spectrograms are used.

Line 171: There are other factors than electron-hole binding that also affect exciton absorption (dipole, electron-hole overlap, state of exciton). How are they included to the conclusion/analysis?

Fig. 4b: Is there an easier way to detect chirality in band structure?

Reply to the Reviewers (manuscript NCOMMS-20-39148)

We thank both Reviewers for their valuable remarks and questions. After careful consideration of each point in the reports, we revised the manuscript and the Supporting Information accordingly. We believe that the manuscript has significantly benefited from the review process and hope that the Reviewers agree.

Reviewer #1

The Reviewer states:

“The results are novel and reliable. Since the manuscript successfully highlights a potential of HHG as an all-optical method for surface-sensitive spectroscopy, the contents would be interesting to the material science community as well as to the photonics community.”

Our response:

We thank the Reviewer for this very positive assessment of our work.

Specific comments by Reviewer #1

1 The Reviewer states:

“Is the demonstrated sensitivity to the inversion symmetry specific to the strong-field regime or does it appear regardless of the excitation regime? The polarization rotation scans (as shown in Fig.2) for varied excitation intensity would provide the answer. Similarly, is the sensitivity to the crystalline chirality specific to the strong-field regime?”

Our response:

We agree with the Reviewer that the harmonic intensity scaling of bi-circular fields might be very insightful, comparable to recent findings obtained with linearly polarized light¹⁻³. In our measurements with linear and bi-circular driving fields (see figures 2, S2 and S3), we observe a higher sensitivity to the inversion symmetry, i.e., a larger modulation depth of the signal, as a function of the crystal angle, for higher harmonic orders. This effect has also been observed in Ref.³ and is attributed to the non-parabolic band dispersion at higher momenta (larger Fourier components).

While the nonlinearity of the generation process (perturbative vs. non-perturbative) certainly also plays an important role in the determination of the crystal symmetry, as shown in Refs.^{1,4-7}, in our case the fields' polarization shape is key for the symmetry sensitivity. The 3-fold field modulation depth is mainly given by the amplitude ratio of the fundamental and the second harmonic field (see also added inset in figure S1a). Our observations show that we are in a regime that allows for the unambiguous determination of the inversion symmetry, and we can link this information to the chiral signature of the respective material.

In our model for the chiral absorption, the field strength is not considered, and we have carefully checked other potential field-dependent excitation channels, e.g., dressed excitons that could contribute to a chiral generation or absorption of harmonics (see Methods section). For the given field strengths and the excitation geometry (near normal incidence illumination), we do not expect any influence from such field-dependent channels, as explained in the Methods section.

We discuss our observations in the revised manuscript in greater detail to highlight the importance of the strong-field regime for the polarization scans and extend the Method section to account for the potential strong-field contributions to the chiral effects.

2 The Reviewer states:

“In Figure 1d, the authors display the polarization rotation scans for H5, H7, H8, and H10. Each order exhibits a three-fold beating, but with different offset phases. Please explain the reason why the offset phase varies with harmonic orders.”

Our response:

We thank the Reviewer for this question. The different offset phases likely stem from the non-perturbative high-harmonic generation, as indicated by studies using linearly-polarized excitation^{4,8-11}. In the real-space picture of high-harmonic generation in solids, the emission phase of a harmonic order is governed by the respective electron-hole trajectories and the birth/recombination time. In future experiments, an analysis in the strong-field picture could complement the reconstruction of real-space electron distributions¹¹ or band dispersions¹⁰.

In the revised manuscript, we explain the observed phase offset in the Results section and extend the statement in line 205 (line 223 in the recent version) to highlight the significance of the phase offset for future experiments.

3 The Reviewer states:

“In the inversion-symmetry probing, does H8 provide any further information than what H5 provides? It is preferable that the authors could address the impact of using higher orders in the structural probing.”

Our response:

We thank the Reviewer for this question, which we partially answered in the response to the Reviewer’s first statement. For odd harmonics, we observe a stronger modulation for higher orders, which is consistent with observation from Ref.³. Since we do not extract any information except the rotational symmetry, for our purposes, H5 would yield the same result as H8 but with a smaller modulation of the signal and, therefore, a smaller angle resolution in the symmetry probing.

We address the impact of higher harmonic orders on the symmetry probing in the revised manuscript.

4 The Reviewer states:

“In addition to the spectral shift shown in Fig.3a and Fig.3b, there are also difference signals for H6, H8, and H10 in Fig. 3a and for H5 and H7 in Fig.3b. What is the origin of these difference signals?”

Our response:

The Reviewer raised a particularly interesting question here. The observed difference signals of the off-resonant harmonics originate from chiral high-harmonic generation efficiencies or dichroic absorption coefficients of the two materials systems under investigation. This observation is consistent with the considered helicity-dependent absorption model.

In particular, we model the chiral dependence of the harmonic absorption by an energy-shifted absorption coefficient $k(E)$, as shown in figure S5. This leads to red and blue shifted on-resonance harmonics but also induces finite chiral absorption differences at energies below and above the resonance.

In the revised manuscript, we now explain the origin of the difference signals of the off-resonance harmonics.

5 The Reviewer states:

“Please provide the angle theta for the data shown in Fig.3c. The difference signal for H7 from quartz (0001) should be angle dependent according to Fig.3a.”

Our response:

We thank the Reviewer for this note. The angle is now provided in the caption of Fig.3. This angle, at which the spectral shift for quartz is maximal, was chosen as an exemplary lineout. The difference signal of the 7th harmonic and also the extracted chiral band shearing in the case of quartz are indeed angle dependent, as shown in Fig.3c and Fig. 5, respectively. This indicates the crystal-orientation-dependent chiral band splitting in quartz.

6 The Reviewer states:

“One can imagine that linear circular dichroism measurement in the ultraviolet range would also detect the chiral properties. Please provide any advantage of the chirality detection based on HHG.”

Our response:

We thank the Reviewer for highlighting the need to elaborate on the advantages of our method. We agree with the Reviewer that under the consideration of dichroic absorption chiral material properties could be extracted using ultraviolet light. However, specifically in the ultraviolet range and near band resonances of transparent materials, nanometric thin samples are required to realize sufficient transmission. Using the reflection geometry, as shown in our approach, would

likely also work in such experiments, but with the disadvantage of a signal with contributions from the real and imaginary part of the refractive index, which makes the analysis much more complex and lowers the chiral absorption contrast. However, more importantly, the lack of suitable light sources for wavelengths below 200 nm with proper polarization control and focusing optics poses additional experimental challenges.

In contrast, our approach circumvents these limitations and difficulties of a pure absorption experiment, since the harmonic radiation is generated in reflection in the crystal itself, and allows for symmetry-related probing of chiral material properties. Additionally, the inherently short (femtosecond) pulse duration as well as the broad spectral bandwidth of high harmonics are important aspects which can be utilized to gain further insights to dynamic electronic and structural properties. This could be particularly relevant for the study of heterostructures.

We extended the conclusion in the revised manuscript to highlight the advantages of our method as a spectroscopic chiral probe.

Reply to Reviewer #2

The Reviewer states:

“The work is an interesting extension of an earlier work, Nat. Photon 11, 227 (2017) by Huber group where linearly polarized light was already shown create distinctive direction and polarization sensitive HH as function of crystal orientation. The connection to this earlier work is neither referenced nor compared to determine the novelty or uniqueness of the present work. Especially, it is not clear which are the truly new aspects proposed in this submission. For example, the conclusion “we demonstrate a unique structural and chiral sensitivity of high harmonics generated from bi-circular laser fields near crystal surfaces” seems an over statement because many aspects of bi-chromatic spectroscopy can also be accessed by other forms of HH spectroscopy. The experiment shown here are obviously high level, but before the unique aspects of bi-chromatic spectroscopy are articulated and demonstrated, the work remains more a technical extension of earlier works. Publication in Nature Communication would require that the authors articulate and demonstrate a unique, important aspect that their approach adds to the HH spectroscopy. In its current form, this manuscript presents more of a technical, incremental extension to prior works”

Our response:

We thank the Reviewer for these comments and for giving us the opportunity to further highlight the novel aspects of our work. The paper by Langer *et al.* is indeed an excellent piece of relevant work, and we acknowledge that deserves explicit mention in our manuscript. We now added it in the context of symmetry detection in solids. We believe that the main difference in our approach is the use of tailored fields possessing discrete rotational symmetry and angular momentum. We demonstrate circularly-polarized high-harmonic generation in inversion symmetric and asymmetric crystals, which constitutes a universally applicable and new method. Furthermore, we use these harmonics to characterize the rotational symmetry *and* chiral properties near a solid material’s surface. Additional novel aspects and consequences of our approach are:

1. The selection rule for HHG suggests a very sensitive probe for dynamic changes of the crystal symmetry, if the conditions for the selection rule are violated by a transiently-changed crystal structure.
2. The gas-free generation of the circularly-polarized coherent light could lead to the development of new compact light sources/converters.
3. The tailored rotational symmetry of the driving field likely maximizes the harmonic yield in fitting crystal symmetries.

In our response to statement 9, we further discuss the novelty and advantages of our approach compared to other surface sensitive spectroscopic techniques.

By considering both Reviewers' comments and suggestions, we hope that the revised manuscript now clarifies the unique aspects of our work, such as the chiral spectroscopic probing. To emphasize that our work goes beyond just symmetry probing, we modified the introduction and the conclusion, including also the first two sentences of the last paragraph, which read now:

“In conclusion, we demonstrate a distinct symmetry-dependent chiral sensitivity of high harmonics generated from bi-circular laser fields near crystal surfaces. We show that the tailored symmetry and angular momentum of the driving field polarization can be utilized to generate circularly-polarized harmonic radiation in solids with arbitrary crystal structures and allow for symmetry-resolved chiral spectroscopy of the generation medium.”

Specific comments by Reviewer #2

1 The Reviewer states:

“Lines 39 & 56: It would be beneficial to elaborate the key idea and consequences of bi-chromatic light fields” with few sentences for a non-expert.”

Our response:

We agree with the Reviewer and modified the introduction and discussion to clarify the key idea and consequences of non-trivial, and specifically bi-chromatic, light fields in a broader context.

In short, we highlight that the rotational symmetry of the field can be tailored by the choice of the constituting excitation light fields. While mixing the fundamental field with its second harmonic yields the three-fold field that we match to three- or six-fold crystals in the current work, equally mixing it with its third harmonic would result in a four-fold light symmetry. In this manner, arbitrary polarization states can be synthesized and tailored to specific inversion symmetries, resulting in new capabilities for more efficient harmonic generation as well as for ultrafast chiral and structural spectroscopy. We now also mention the use of bi-circular solid HHG as a gas-free source for circularly-polarized coherent radiation, which could be relevant for electron-based ultrahigh vacuum experiments.

2 The Reviewer states:

“Lines 80-82: Statement “The near-perfect suppression of harmonic orders $q = 3n$ (3,6,9,...) testifies to the circular polarization of the unsuppressed harmonics” is unclear for a non-expert.”

Our response:

We acknowledge that this statement warrants some more details. We changed the sentence in the revised manuscript and now also refer to the supplementary material, where this topic is discussed in depth.

3 The Reviewer states:

“Fig 1a: What is the role of field strengths (besides the phase delay) of two pulses to create the desired bi-circular field?”

Our response:

The amplitude ratio between fundamental and second harmonic determines the shape, i.e., the symmetry modulation, of the resulting driving field but not its rotational symmetry. We now added an inset in figure S1a showing driving fields for the amplitude ratios of 1:1, 1:2 and 1:5. As mentioned in the response to the first comment by Reviewer #1, the specific field shape should influence the sensitivity, i.e., the resolution, of the rotation scans. However, symmetry aspects like selection rules and the generation of circular harmonics are unaffected.

We added a statement at the end of the section “inversion-symmetry probing” and modified the Methods section in the revised manuscript for clarification.

4 The Reviewer states:

“Figs 1d & 2: How unique are the crystal-direction dependencies to bi-circular approach? It seems many of the key features reported can be extracted by analyzing the details of a linear excitation. This approach is much simpler than bi-circular, and it has been demonstrated before. Or is the message that one should use few excitation schemes to resolve crystal symmetries? If yes, can you propose a combination of excitation schemes that works to characterize most crystals?”

Our response:

This is an important issue, and we think it is necessary to discuss the uniqueness of the crystal-direction dependencies in the broader context of our approach.

As indicated by the Reviewer, other techniques using linear light polarizations can be used to determine crystal symmetries. However, these techniques rely on a polarization-angle scan in combination with a harmonic polarization state analysis to unequivocally determine the correct point group. As shown in our work, the three-fold structure of quartz is accurately determined with the three-fold field by using a polarization scan of the driving field only. While this in

itself represents a unique feature of the bi-chromatic HHG, it is just one part of the overall message we are trying to convey here. The sensitivity to the crystal symmetry fully plays out in combination with the chiral spectroscopy, which we view as the key finding of our study. The inversion-asymmetric polarization of the three-fold field allows for the mapping of chiral features along crystal axes, which we exemplify for the structural chirality in quartz. On the other hand, the absence of such a directionality indicates isotropic chiral features, such as the surface magnetism of MgO. In essence, it is the combination of crystal-direction dependencies with chiral probing that poses a great opportunity for studying chiral materials systems.

We revised the introduction and discussion of the manuscript for clarification.

5 The Reviewer states:

“Line 119: Statement “A linearly-polarized field has an equal amplitude under inversion, with an opposite sign” is correct but probably will not open to a non-expert.”

Our response:

We agree with the Reviewer that this phrasing can be improved, and we revised the text accordingly.

6 The Reviewer states:

“Line 127: Why is the 12-fold symmetry not resolvable by the used setup?”

Our response:

As discussed in comment 3, the amplitude ratio of the bi-chromatic field influences the shape of the three-fold field and therefore the modulation in the rotation scans. The ratio of 1:5 for the scans of MgO results in a less pronounced three-fold modulation of the field and leads to a worse angular resolution in rotation scans compared to measurements on quartz (ratio of 1:2). Additionally we find that the angle resolution in 4-fold MgO(100) is intrinsically worse compared to 6-fold MgO (111), which stems from a larger isotropy in the HHG efficiency in MgO(100) (see Figs. 2e and f).

We added this explanation to the revised manuscript.

7 The Reviewer states:

“Line 145: Caption text “Difference signal of two intensity-normalized spectrograms” does not specify clearly which two spectrograms are used.”

Our response:

We thank the Reviewer for this comment. The revised manuscript now specifies that we used two intensity normalized spectrograms for which the helicities of the fundamental and second harmonic have been flipped. Thereby, the helicity of all harmonics is inverted such that the difference spectrogram traces the dichroic signal of the HHG and the subsequent absorption.

8 The Reviewer states:

“Line 171: There are other factors than electron-hole binding that also affect exciton absorption (dipole, electron-hole overlap, state of exciton). How are they included to the conclusion/analysis?”

Our response:

We thank the Reviewer for this question. In our analysis of the chiral response, we did not assume any microscopic mechanism but rather fitted the absorption change of an absorption edge shift with an arbitrary origin. In quartz, our data is best described with a chiral absorption taking place in the whole probed volume. However, in MgO we found an effective chiral interaction length of 6.5 nm that is comparable to the exciton diameter. This value is large compared to the extent of the magnetic surface layer that is expected in this material. Our hypothesis is that the relevant absorption near the band gap is mediated through an excitonic state that remains constant under an inverted helicity of the driving field (here we exclude an optical stark effect as explained in the Methods section). Hence, the spin polarized bands at the surface of MgO may lead to an exciton mitigated chiral absorption in an extended region on the order of the exciton diameter.

9 The Reviewer states:

“Fig. 4b: Is there an easier way to detect chirality in band structure?”

Our response:

This question also relates to statement 6 from Reviewer #1. We again thank both Reviewers for their comments regarding the general context of chiral spectroscopy.

As we discussed in detail in our response to Reviewer #1, linear or circular dichroism absorption experiments could be utilized to access chiral band structure information. However, the requirements on suitable light sources with polarization control, focusing optics and thin samples are rather limiting factors in such experiments.

There are other techniques, such as spin-resolved photoemission spectroscopy or spin-polarized scanning tunneling microscopy, which can provide access to chiral electronic signatures. Both of these techniques are based on electron probes, requiring ultrahigh vacuum conditions and atomically clean surface terminations. Specifically for insulating media, such electron-based, surface-sensitive techniques are experimentally challenging, since surface charge and surface contamination over time are major issues on non-conductive surfaces¹².

Chiral spectroscopy with solid-state HHG is an all-optical approach that can be tailored to several material systems and has relaxed experimental requirements on the vacuum conditions and sample preparation. We believe that this method constitutes a particularly interesting and promising route for chiral band structure probing and extends the rich toolbox of solid-state ultrafast spectroscopy.

We highlight these advantages in the conclusion of the revised manuscript.

Literature:

1. You, Y. S., Reis, D. A. & Ghimire, S. Anisotropic high-harmonic generation in bulk crystals. *Nature Phys* **13**, 345–349 (2017).
2. You, Y. S., Lu, J., Cunningham, E. F., Roedel, C. & Ghimire, S. Crystal orientation-dependent polarization state of high-order harmonics. *Opt. Lett.*, *OL* **44**, 530–533 (2019).
3. Liu, H. *et al.* High-harmonic generation from an atomically thin semiconductor. *Nature Physics* **13**, 262–265 (2017).
4. Luu, T. T. & Wörner, H. J. Observing broken inversion symmetry in solids using two-color high-order harmonic spectroscopy. *Phys. Rev. A* **98**, 041802 (2018).
5. Langer, F. *et al.* Symmetry-controlled temporal structure of high-harmonic carrier fields from a bulk crystal. *Nature Photonics* **11**, 227–231 (2017).
6. Luu, T. T. *et al.* Extreme ultraviolet high-harmonic spectroscopy of solids. *Nature* **521**, 498–502 (2015).
7. Kumar, N. *et al.* Second harmonic microscopy of monolayer MoS₂. *Phys. Rev. B* **87**, 161403 (2013).
8. Luu, T. T. & Wörner, H. J. High-order harmonic generation in solids: A unifying approach. *Phys. Rev. B* **94**, 115164 (2016).
9. Luu, T. T. & Wörner, H. J. Measurement of the Berry curvature of solids using high-harmonic spectroscopy. *Nature Communications* **9**, 916 (2018).
10. Vampa, G. *et al.* All-Optical Reconstruction of Crystal Band Structure. *Phys. Rev. Lett.* **115**, 193603 (2015).
11. Lakhotia, H. *et al.* Laser picoscopy of valence electrons in solids. *Nature* **583**, 55–59 (2020).
12. Woodruff, D. P. Quantitative Structural Studies Of Corundum and Rocksalt Oxide Surfaces. *Chem. Rev.* **113**, 3863–3886 (2013).

Reviewers' Comments:

Reviewer #1:

Remarks to the Author:

In this manuscript, the authors demonstrate a structural/chiral sensitivity of HHG at solid surfaces. The detection of inversion symmetry and the probing of the crystalline chirality are demonstrated with HHG driven by bi-circular laser fields. The results are novel and reliable. Since the manuscript successfully highlights a potential of HHG as an all-optical method for surface-sensitive spectroscopy, the contents would be interesting to the material science community as well as to the photonics community. The manuscript has been revised in an appropriate manner according to the comments raised by the reviewers, the manuscript can now be accepted for publication.

Reply to the Reviewers (manuscript NCOMMS-20-39148)

Reviewer #1

The Reviewer states:

“In this manuscript, the authors demonstrate a structural/chiral sensitivity of HHG at solid surfaces. The detection of inversion symmetry and the probing of the crystalline chirality are demonstrated with HHG driven by bi-circular laser fields. The results are novel and reliable. Since the manuscript successfully highlights a potential of HHG as an all-optical method for surface-sensitive spectroscopy, the contents would be interesting to the material science community as well as to the photonics community. The manuscript has been revised in an appropriate manner according to the comments raised by the reviewers, the manuscript can now be accepted for publication.”

Our response:

We thank the reviewer again for the positive assessment of our work and the recommendation to accept the manuscript for publication.

Specific comments by Reviewer #1

1 The Reviewer states:

“The response to the question on 'whether the demonstrated sensitivity to the inversion symmetry and to the chiral properties specific to the strong-field regime or not' is not very convincing. A larger modulation depth of the signal for higher harmonic orders would happen not only for non-perturbative HHG but also for the perturbative HHG. If the sensitivity is not specific to the strong-field regime, it should be stated.”

Our response:

We agree with the reviewer, that a sensitivity in the perturbative regime cannot be excluded. We modified our statement on the modulation depth of the signal to reflect, that a perturbative mechanism could yield the same effect.

2 The Reviewer states:

“Even if we adopt that the emission phase of a harmonic order is governed by the respective electron-hole trajectories and the birth/recombination time, it is not trivial why this emission phase causes the observed offset phases of the polarization rotation scans.”

Our response:

We agree that the offset phase is a non-trivial result of the generation mechanism. In this study we did not analyze the mechanism behind the emission phase and suggest further work in this direction. We modified the sentence in our manuscript that highlights the potential of further work on the emission phase for investigations of various solid state phenomena.